# Development Priorities for the Regional Innovation System Based on the Best Available Technologies

**Nikolay Kuznetsov [1], Sergey Tyaglov [1], Marina Ponomareva [1], Nataliya Rodionova [1] and Karina Sapegina [2,\*]**

[1] Rostov State University of Economics, St. B. Sadovaya, 69, 344002 Rostov-on-Don, Russia; kuznecov@rsue.ru (N.K.); tyaglov-sg@rambler.ru (S.T.); yuma@list.ru (M.P.); ndrodionova@mail.ru (N.R.)
[2] Peter the Great St. Petersburg Polytechnic University, Polytechnicheskaya St., 29, 195251 St. Petersburg, Russia
\* Correspondence: sapegina.k@edu.spbstu.ru

**Abstract:** At the present stage one of the most important factors in the economic growth of Russian regions is the production processes modernization based on the best available technologies (hereinafter—BAT), ensuring the reduction of the negative impact on the environment in cost-effective ways. The most important conditions for the successful implementation of BAT at regional enterprises is the creation of general institutional conditions at the all-Russian level and the supply of enterprises with domestic technologies meeting the criteria of the best available technology. Over the past several years, large-scale work has been carried out in Russia to amend legislation, prepare BAT reference documents, and form the institutional conditions for their implementation. However, sustainable outcomes in specific regions will require the further development of regional innovation systems, consistent with the needs of local enterprises in BAT. The article proposes a general regional mechanism for managing the sustainable development of an innovation system based on BAT. For its successful implementation, a set of practical recommendations for the Rostov region has been formed. Within the framework of the innovation regional infrastructure it was proposed to create a new institute for the implementation of BAT—the Regional BAT competence center, priority areas for the development of potential and promising BAT in the region were identified, a general pattern of interaction of the Regional BAT competence center with other participants in the process was developed.

**Keywords:** regional innovation system; best available technologies; sustainable development; competitiveness

## 1. Introduction

Best Available Technologies are "technologies and arrangements that minimize the impact on the environment as a whole and that are not costly to implement" [1].

In the countries of the European Union, this tool, combined with integrated environmental permits and BAT Guides, is one of the most effective ways to reduce the negative impact of production activities on the environment in cost-effective ways.

At the same time, in non-European countries, the application of the BAT-based approach is complicated by the technical incompatibility of European and non-European technologies and technological processes, as well as the different levels of financial capabilities of European and non-European enterprises.

In this regard, there is a sufficient number of papers by foreign and Russian scientists covering various aspects of ways to solve this practical problem of BAT implementation.

In particular, the first way is the adaptation of European BATs to the specific conditions of particular countries.

Within the framework of this direction, a number of works are devoted to the development of methods for determining and selecting the best available technologies for the purposes and adaptation of European technologies to the economies of non-European countries.

For example, Roger Dijkmans proposes multi-stage procedure of the BATs sectorial selection, which allows assessing the technologies in terms of technological feasibility,

economic efficiency and ecological parameters that influence the environment and its components [2].

The first step in the procedure proposed by Roger Dijkmans is evaluation of candidate's technologies for BAT status in terms of their technical capability [2]. This allows us to immediately weed out all existing technologies, the practical implementation of which in this industry in this country isn't possible. It also guarantees the technological compatibility of technologies that received BAT status as a result of selection with technologies from non-European countries.

However, this does not guarantee the mandatory availability of compatible technologies in the event that the introduction of all considered European candidate technologies turns out to be technically impossible. If this situation turns out to be real, technologies of local origin will be required, taking into account the specifics of the development of a particular industry in a particular country. Then, on the whole, the approach proposed by the author will work only if there are local candidate technologies that compete with each other and with the corresponding European technologies. In addition, the expert nature of the proposed assessment, a lot of will depend on the composition and qualifications of experts making a decision to include the technology in the BAT, which may lead to ineffective decisions at first.

Schollenberger H., Treitz M., Geldermann J. consider the necessity to adapt the BAT transfer technology into the industrially developed countries, in particular, to identify the main criteria due to the differences in economic, legal and technical conditions [3].

The authors note that by reason of these differences, technologies recognized as BAT in European economies won't necessarily match BAT requirements in emerging economies, and vice versa. Therefore, they have developed a flexible approach to defining BAT, taking into account regional characteristics and preferences of individual countries. As in the previous case, the proposed method includes different criteria and combines technological, economic and environmental parameters. Specific national conditions can be taken into account through the use of individual emission limit values as well as country-specific criterion weights from which the country-specific BAT metrics are determined.

In fact, like the previous authors, Schollenberger H., Treitz M., Geldermann J. tried to build a decision-making algorithm to assess the possibilities of transferring environmentally friendly technology for use in a specific country. The presented method for determining which country to use BAT can be applied by political decision makers and local authorities for accrediting BAT, as well as by technology providers.

At the same time, its practical application is complicated in determining the weight coefficients which used to take into account the regional and national specifics of the possibilities of introducing foreign BAT.

The work by Mavrotas G., Georgopoulou E., Mirasgedis S., Sarafidis Y., Lalas D., Hontou V., Gakis N. is devoted to adapting and technological compatibility of the foreign best available technologies while introducing them into the developing countries [4].

Sectoral aspects of the BAT introduction are considered in the article by L. Loyon, C.H. Burton, T. Misselbrook, J. Webb, F.X. Philippe, M. Aguilar, M. Doreau, M. Hassouna, T. Veldkamp, J.Y. Dourmad, A. Bonmati, E. Grimm, S.G. Sommer [5] (on the example of animal husbandry). The authors note the insufficient completeness and inconsistency of BAT in the field of animal husbandry (in particular, pig and poultry farming), united within the EU. There are highlighted such problems in terms of their adaptation in other countries, such as the complexity of regulation, as well as counterproductiveness in relation to other goals (for example, the exchange of pollution). This requires a comprehensive approach to evaluating different BAT methods to develop effective farm-specific strategies.

The authors also conclude that the achievement of a significant result for the environment in animal husbandry based on European BAT in other countries is associated with significant costs for farmers and a decrease in the economic results of farms, and does not take into account the conditions of farming in these countries. In this connection, the interest of farmers in their implementation requires additional incentives.

Thus, in general, the adaptation of European BAT into the practical activities of enterprises and farms in other countries is associated with significant difficulties.

These difficulties are also associated with direct decision-making by specific enterprises and regulatory bodies of these countries on the introduction of certain BAT in the production process.

In this regard, another way to adapt foreign BAT is to develop tools to justify the introduction of BAT at the level of enterprises and regulatory authorities.

For example, A. Cikankowitz, V. Laforest in their article propose the methodology to assist the manufacturers, authorities while implementing the enterprise operating license renewal procedures focused on evaluating the existing BAT methods and ensuring the basis for assessment of production units and management processes [6].

Damien Evrard, Jonathan Villot, Chadad Armiyaou, Rodolphe Gaucher, Sofia Bouhrizi, Valerie Laforest propose the integrated method to multi-criteria assessment of the reference installations to introduce the BAT based on mathematical and statistical methods [7,8].

The article by D. Huybrechts, A. Derden, L. Van den Abeele, S. Vander Aa., T. Smets analyzes the possibility of implementing the principles of sustainable supply chain management as the basis for the definition of BAT, which can become a driving force for the ecology of global value chains [9].

The process of introducing BAT in Russia is also widely studied by Russian scientists. Here we should highlight the article by T.V. Nevalenov, O.N. Lazdin, which analyzes the role of the voluntary environmental certification system in the process of accelerating the introduction of the best available technologies [10]. Also, T. V. Guseva, Ya. P. Molchanova, M.V. Begak, A.V. Mironov pay special attention to the development of energy management systems in Russia, as well as to the problem of personnel training to ensure the transition to BAT [11].

Many Russian scientists—T.V. Guseva, Y. P. Molchanova, M. V. Begak, K. A. Shchelchkov, O.V. Grevtsov, N.V. Kostyleva, I.S. Poddubny, I.I. Rebrik, A.G. Bernyatsky, R.V. Starshinov, I.A. Kosorukov, S.A. Konstantinova, V.B. Sapozhnikov, etc. [12] also consider the introduction of BAT in various sectors of the Russian economy—nickel industry, lead production, precious metals, mining, etc.

Thus, the adaptation of foreign technologies to specific national systems is always associated with a significant difficulties described above. The second, and in the opinion of the authors of this article, more beneficial for non-European countries, the direction of introducing BAT in them is to stimulate the subjects of national (local, regional) innovation systems for the development of potential, promising BAT and BAT itself. In this case, they will initially correspond to all technical, economic and environmental conditions of specific national economies and their subjects. This direction is practically not represented in the scientific literature.

It should be borne in mind that the introduction of BAT at enterprises is not only the greening of production.

For countries and regions in general, in particular for the Rostov region, these are opportunities for restructuring the economy and increasing the share of industries with high added value through the modernization and expansion of production, the transition to energy-efficient technologies, improving conditions and increasing labor productivity, increasing the competitiveness of enterprises. region. And in the long term—the possibility of entering new domestic and foreign markets for innovative environmentally friendly technologies in the field of potential BAT and strengthening the innovative sector of the economy. All these are the characteristics of development. Investment in these spheres will provide a significant multiplier effect for the economy and social sphere of the region in the future. Regional authorities and investors may be highly interested in motivating enterprises to participate in the development and implementation of potential and promising BATs [13].

First of all, the level of institutional conditions for the implementation of BAT at the federal level is extremely important for the regions. In recent years, large-scale work

much has been done in Russia to create a full-fledged institutional environment for the implementation of BAT.

The list of areas of BAT application was approved, criteria were determined and methodological recommendations were approved for defining technology as the best available technology, a clear legislative consolidation of this term was ensured, measures were envisaged to stimulate enterprises to implement BAT [13].

Since 2019, the national project "Ecology" has been implemented in Russia and its regions, within the framework of which measures are provided for the implementation of BAT. By the start of the project the Bureau of the best available technologies [14] (www. burondt.ru accessed on 16 October 2021) had been formed [15].

Also on the basis of the Federal Agency for Technical Regulation and Metrology (abbreviated as Rosstandart)—the federal executive authority that performs functions for the provision of public services, state property management in the field of technical regulation, standardization and ensuring the uniformity of measurements [16] Rosstandart has a specialized technical committee for standardization "The best available technologies".

Rosstandart is a base for a specialized technical committee for standardization of "Best available technologies". For the development of information and technical reference books, technical working groups have been formed (BAT Bureau website). Rosstandart approved and published BAT reference books [17,18]. Currently, there are already positive examples of such implementation at Russian industrial enterprises [19].

In addition, the most important condition for the successful implementation of BAT at enterprises in the regions is to provide them with domestic technologies developed on a scale necessary to meet the criteria of the best available technology.

In this regard, a mechanism for the development of innovative systems based on BAT should be launched at the regional level, the aim of which is to synchronize the functioning ofboth innovative institutions and institutions for regulating the rational use of natural resources, which currently operate in a relatively isolated manner. The general outline of such a mechanism with a detailed description is presented by the authors in a number of previous works [20–23]. Its main result is the expansion of the potential and promising BAT flow in the region for their subsequent introduction to the relevant reference books on BAT and their implementation at the regional enterprises, through the development of interaction of all participants in the process of creating and implementing such technologies and providing a system of incentives.

The general description of the mechanism as a whole can be applied to any Russian region. However, its practical implementation in a specific region will objectively have its own specifics, due to the peculiarities of the existing innovation infrastructure, the specialization of the innovation sector, measures to support investors and innovators developed in the region, as well as the structure of the production sector, which forms the demand for BAT.

In this regard, the purpose of this article is to develop additional practical recommendations for the Rostov region aimed at implementing a mechanism for the development of a regional innovation system based on BAT. In this regard, the Rostov region needs additional practical recommendations aimed at implementing the proposed mechanism for the development of a regional innovation system based on BAT. Recommendations for additional support for enterprises implementing BAT at the regional level, creating additional preferential instruments to attract private investment for research and development activities in the field of creating potential and promising BAT in the region, separately for small and medium-sized enterprises, the introduction of these measures and tools to the current system of business and investor support institutions in the region were presented in a number of previous articles by the authors [13,21,22].

In this article, in order to achieve the above research goal, the following tasks will be solved (research plan):

—provide a general description of the proposed mechanism for the development of a regional innovation system based on BAT;

—define the concepts and give a comparative characteristic of the actual BAT, potential and prospective BAT;

—to form recommendations on the creation of a new institute for the introduction of BAT in the Rostov region—the Regional Center of BAT Competencies (including general methodological recommendations on its structure, the scheme of interaction of specialists of the Regional Center of BAT Competencies with other participants in the process and the identification of the most promising areas for the development of potential and promising BAT in the region);

—to develop the basics of the program for the implementation of the RCC on BAT in the region;

—to determine its possible impact on the main indicators of innovative development of the Rostov region and to identify priority areas of specialization of the region for the development of potential BAT and subsequent implementation of BAT.

In this article the authors focus on the recommendations for the creation of a new institution for the implementation of BAT in the Rostov region—the Regional BAT competence center, including general methodological recommendations on its structure, the patterns of interaction between the specialists of the Regional BAT competence center with other participants and identification of the most promising areas for development of potential and promising BAT in the region. These recommendations were created taking into account the needs of Rostov region enterprises in BAT, as well as the participation of the Rostov region in the development of the world-class Southern Research and Educational Center.

## 2. Analysis of Works and Problem Statement

Analysis of the scientific literature on the issues under consideration shows that there is a sufficient number of scientific studies by Russian and foreign scientists devoted to the certain components of the mechanism proposed in the article: both in the field of innovation development, stimulation of innovation and the formation of effective innovative institutions, in the field of environmentally friendly, energy efficient technologies, on the one hand, and in terms of improving the mechanism for introducing BAT in various countries, including procedures for their selection, solving problems of compatibility of foreign technologies with national technological systems, on the other hand.

Among the scientific publications on the development of innovative activities, the work of Smol M., Kulczycka J. [24], devoted to the problem of the development of innovations in the European raw materials sector, should be highlighted. Palage K., Lundmark R., Söderholm P. examine the innovative effects of renewable energy policies and their interactions (using the solar photovoltaic industry as an example) [25]. Pitelis A.T., Vasilakos N., Chalvatzis K., Pitelis C.N. analyze the possibilities of industrial policy to stimulate the transfer of technological innovation in the field of renewable energy to the OECD and to the regions of the European Union [26]. Zemtsov, S.P., Chernov, A.V. focus on the reasons for the different growth rates of different high-tech companies in Russia [27].

The financial aspects of turning the innovations into clean energy production in OECD countries are analyzed in the article by Al Mamun M., Sohag K., Shahbaz M., Hammoudeh S. [28]. The regional aspect of the development of innovations and the effectiveness of indirect support of innovations in different regions is mentioned by Klímová V. [29]. An article by Böhringer C., Cuntz, A., Harhoff D., Asane-Otoo E. [30] is devoted to the analysis of econometric methods for studying the impact of climate policy on the development of renewable energy sources.

On the other hand, there are quite a few articles devoted specifically to the problem of BAT implementation:

—sectoral aspects of BAT implementation, including stimulation and preparation of BAT reference documents, are considered by a number of Russian and foreign scientists: Dijkmans R. [2], L. Loyon, C.H. Burton, T. Misselbrook, J. Webb, F.X. Philippe, M. Aguilar, M. Doreau, M. Hassouna, T. Veldkamp, J.Y. Dourmad, A. Bonmati, E. Grimm, S.G. Sommer [5], Akimova N., Begak M., Voronina Z., Guseva T., Rebrik I., Toshchev D., I. Ilyukhin,

Tarasov A., Serdyukov V., Konstantinova S., Sapozhnikov V., Poddubny I., Kostyleva N., Molchanova Y. [31], Zakondyrin A. [32,33], Garbarino E., Orveillon G., Saveyn Hans GM 161 [34], C. Magomedov, I. Guseva [35], and others;

—works devoted to decision-making on the implementation of BAT at the level of enterprises and regulatory authorities are presented by such scholars as Cikankowitz A., Laforest V. [6], Evrard D., Villot J., Armiyaou C., Gaucher R, Bouhrizi S., Laforest V. [7], Huybrechts D., Derden A., L. Van den Abeele, S. Vander Aa., Smets T. [9]; Terent'eva Z. [36], Sergey N. Volkov, Olga N. Rublevskaya, Irina O. Tikhonova, Tatiana V. Guseva, Matti Iikkanen [37];

—issues of adaptation and technological compatibility of foreign best available technologies at the stage of their implementation in developing countries H. Schollenberger, M. Treitz, J. Geldermann [3], G. Mavrotas, E. Georgopoulou, S. Mirasgedis, Y. Sarafidis, D. Lalas, V. Hontou, N. Gakis [4], Shaydurova A.A. [38] and others;

—in addition to the above areas, the issues of determining the relationship of the BAT institute with other institutions of rational environmental management remain valid for Russian researchers (T.V. Nevelenova, O.N. Lazdina [10]), as well as the creation of an institutional environment for the implementation of BAT as a tool for environmental activities (O. P. Burmatova [39]), considering the best available technologies as a tool of industrial and environmental policy (T.V. Guseva, Ya.P. Molchanova, M.V.Begak, A.V. Mironov [11]), their relation to the modernization of the Russian economy, the transition to the model of "green economy" (L.A. Mochalova [40], Yu.A. Timofeeva [41], Ovchinnikova I.N. [42], Dovbiy IP, Degterenko AN, Kobilyakova V.V. [43]), institutional aspects of BAT implementation under Russian conditions [39]), tasks of personnel training in the field of technological regulation of industry based on the BAT principles for Russian and international projects [40], financial aspects of BAT implementation—Prosvirina I.I., Dovbiy N.S. [43–46], etc. [47–51].

At the same time, despite a fairly large amount of publications concerning the abovementioned issues, there is little research on combining and synchronizing the activities of innovation systems' subjects and environmental management systems' subjects in regulating the implementation of BAT (both on state and regional levels). There are no applications for specific Russian regions, including for the Rostov region.

Therefore, research to create practical recommendations for the development of innovation systems in Russian regions based on the best available technologies would without a doubt be useful.

Applications for specific regions to create new institutions for BAT implementation, adapted to the specifics and structure of their economy, the actual need of enterprises for BAT and the potential of innovation subjects, in particular for the Rostov region, are of great importance.

The lack of concrete results in this sphere in terms of practical use by Russian regions made us focus our research on making recommendations for the development of innovative systems based on the best available technologies. Specific regional applications on creating new institutions for BAT implementation, taking into account the specifics and structure of their economy, the actual need of enterprises for BAT and the potential of subjects of innovation, for the Rostov region in particular has assumed a particular urgency.

## 3. Solution Method

As research methods general scientific methods, including observation and abstraction, deduction and induction, economic, logical and comparative analysis, generalization of factual and theoretical material were used.

When identifying priority areas for the development of potential and promising BAT in the region (in the Rostov region), methods of grouping data, normative and systemic methods, methods of tabular and graphical data visualization were used.

Proposals for the creation of new institutes for the implementation of BAT in the Rostov region (Regional BAT Competence Center, BAT Process Factory) were formed using the method of analogies. As a basis for the analogy, the region's experience in the creation

of similar institutions in the field of labor productivity, ensuring the introduction of lean technologies tools at the enterprises of the region, is used. In general, both BAT and lean technologies are aimed at increasing production efficiency and productivity of labor and resources, but in different ways. In this connection, the general methodological approaches to these institutes' activities (the availability of expert professionals accompanying the process of introducing technology at enterprises, their selection, personnel training, free of charge services provided by the center and the BAT Process Factory, etc.) can be adopted to the BAT implementation. At the same time, when drawing an analogy, it is necessary to take into account that in contrast to lean technologies, the introduction of BAT into the production process may a major investment and approval of regulatory authorities [13]. In addition, it should be emphasized that the analogy in this case applies only to the ways in which the experts of the RCC on BAT interact with enterprises and innovative organizations that will be involved in the process of developing and implementing BAT. The analogy is also applicable in terms of the principles of the formation of the RCC staff, as well as the financing of its activities and the services of RCC experts completely free of charge for enterprises (experts are included in the staff as permanent employees at the main place of work). In terms of the specifics of the technologies being implemented, the analogy between lean manufacturing technologies and BAT is inapplicable, since the latter differ significantly. In this regard, the experts themselves, their qualifications, and field of activity will be subject to requirements that differ from lean manufacturing experts.

When forming the general scheme of interaction between the Regional BAT Competence Center and other participants of BAT introducing processes at regional enterprises, as well as developing potential and promising BATs, a systematic approach was used, which makes it possible to highlight the main elements and participants of a common system of interaction, as well as to determine the relations between them.

When determining the effect of the introduction into the practice of regional management of a mechanism based on the creation of the RCC for BAT, a program-target approach was used, including the formation of the main indicators of a possible program for the introduction of BAT in the region, their target values, and determining their impact on the overall indicators of innovative development of the region.

Public materials of the Federal State Statistics Service of the Russian Federation and its Rostov regional subdivision, documents of the Ministry of Science and Higher Education on innovation infrastructure and the main indicators of innovative activity of the constituent entities of the Russian Federation, including the Rostov region, public records of official web portal of the executive authorities of the Russian Federation and Rostov region, as well as scientific articles by Russian and foreign economists on the issues under consideration, and longstanding scientific and practical experience and generalizations of the authors are the information and empirical basis of the materials and findings presented in the article.

The regulatory framework of the study is based on the regulatory legal acts of the Russian Federation, its regions, regulatory legal acts of the Rostov region, programs, guidelines and policy documents concerning the regulation of innovative activities and the introduction of BAT in the Russian Federation and regions.

## 4. Results

The general theoretical basis of the mechanism for managing the sustainable development of regional innovation systems based on the best available technologies was presented in a number of previous articles by the authors [21–23].

A brief description is given here for a better understanding of further recommendations. The proposed mechanism is built at the junction of two systems currently operating in Russian regions which are relatively isolated from each other—that is regulation of rational environmental management based on BAT (as a sector where BAT is being introduced), and regional innovation systems as a sector where potential/promising BAT and their first approbation and implementation is taking place (Figure 1).

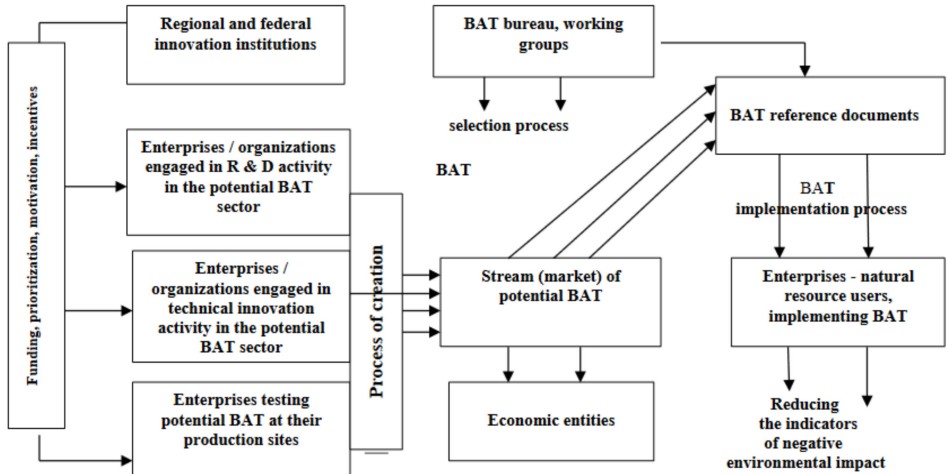

**Figure 1.** Outline representation of the mechanism for managing sustainable development of regional innovation systems based on BAT.

One of the key elements of the mechanism is BAT Reference Books, which are formed on the basis of technologies that at the moment have the best effects among the scientific and technological advances in terms of environmental impact indicators.

It is also extremely important that the implementation of BAT should be carried out at a financial level affordable for the enterprise. BAT has mandatory parameters of economic efficiency, which should also be the best. It allowed for the improvement of both environmental effective and economic efficient technologies. Thus, the mechanism for regulating the rational use of natural resources based on BAT and their reference books actually regulates their consumption sector, where direct demand is generated by enterprises, users of natural resources [13].

According to the diagram shown in Figure 1, regional innovation systems become "suppliers" of BAT in this case. The main goal of the proposed mechanism creation is to increase the number of potential BAT of domestic origin and to form a technological basis for constant modernization and greening of economic activities, and as a result, to reduce the overall negative impact on the environment [13].

The concept of a "potential BAT" was introduced for a more accurate understanding of the process such a technology "moves" from the point of the first research and development to its direct creation, experimental implementation, testing and to its recognition as the "best available" and already included in the BAT reference book, i.e., to justify the connection between the two segments of the mechanism presented in Figure 1.

A potential BAT is understood as an innovative technology that has BAT properties (improved result in terms of environmental impact parameters in the industry; economic efficiency; financial affordability; the possibility of production scaling), claiming to be included in the relevant industry BAT reference book in the future, and being at various stages of innovative development (from research to implementation) [13]. The introduction of this concept was required to define BAT as a priority research and development sphere for innovative enterprises and regional development institutions. It is the flow of such potential BAT that should be increased so that there would be competition between them, and in the end one of them would become the best.

The concept of "promising technologies" has come into use in Russian BAT implementation practice originated. It stands for technologies that have not yet been widely spread, and there is no evidence of industrial implementation of which at two facilities (enterprises) operating in the Russian Federation. In this case, the technology is recommended to be included in the list of promising technologies. As a rule, such technologies include technologies in relation to which research and development work is still being carried out or their experimental-industrial implementation is on. This concept, being essentially close to the "potential BAT", nevertheless has a significant difference—promising

technologies are already included in the list of technologies, and when introduced at two Russian enterprises, they will be able to graduate to the BAT category status. Potential BAT are technologies at any stage of development, just aiming to be included in the list of promising technologies or BAT reference books [13].

Comparative characteristics of these types of BAT aiming at a more accurate understanding of their role in the mechanism for managing the sustainable development of regional innovation systems based on BAT are presented in Table 1.

**Table 1.** Comparative characteristics of "potential", "promising" BAT and BAT.

| № п/п | Comparison Criteria | Potential BAT | Promising BAT | BAT |
|---|---|---|---|---|
| 1 | Stage of creating an innovative product | From fundamental research to the first R&D and implementation stages | Technologies that have certificates of industrial implementation at less than two enterprises in the Russian Federation | Industrial implementation at two or more enterprises in the Russian Federation |
| 2 | Environmental impact indicators | Improved compared to those included in BAT reference books | | The best to date |
| 3 | Economic and financial indicators | Improved compared to those included in BAT reference books | | The best to date |
| 4 | Amount of development projects, extended practice | Can be developed in parallel by several innovative organizations with different parameters, form a stream of market proposals for potential implementation | Implementation with a validated result at least at 2 enterprises is necessary for each technology in order to graduate to the BAT category status | only one BAT in its sector at the moment is selected |

Thus, by introducing the specified types of BAT, it is possible to ensure the link between the innovation sector itself and the BAT implementation sector.

However, for the effective implementation of the mechanism proposed, it is necessary to provide institutional support for the interaction of all participants in the development and implementation of all types of BAT in the region, as well as their subsequent possible implementation at enterprises in other regions as well as at foreign enterprises in the future.

Concerning Russian regions, including the Rostov region, such institutional support can be implemented through the creation of a specialized structure—the Regional Center of Competence in BAT (hereinafter—RCC in BAT), which can be established on the basis of one of the existing institutions (for example, on the basis of the Innovation Agency of the Rostov Region).

The place of the RCC in BAT in the general scheme of interaction of participants in the process of introducing various types of BAT in the Rostov region can be schematically represented as follows (Figure 2).

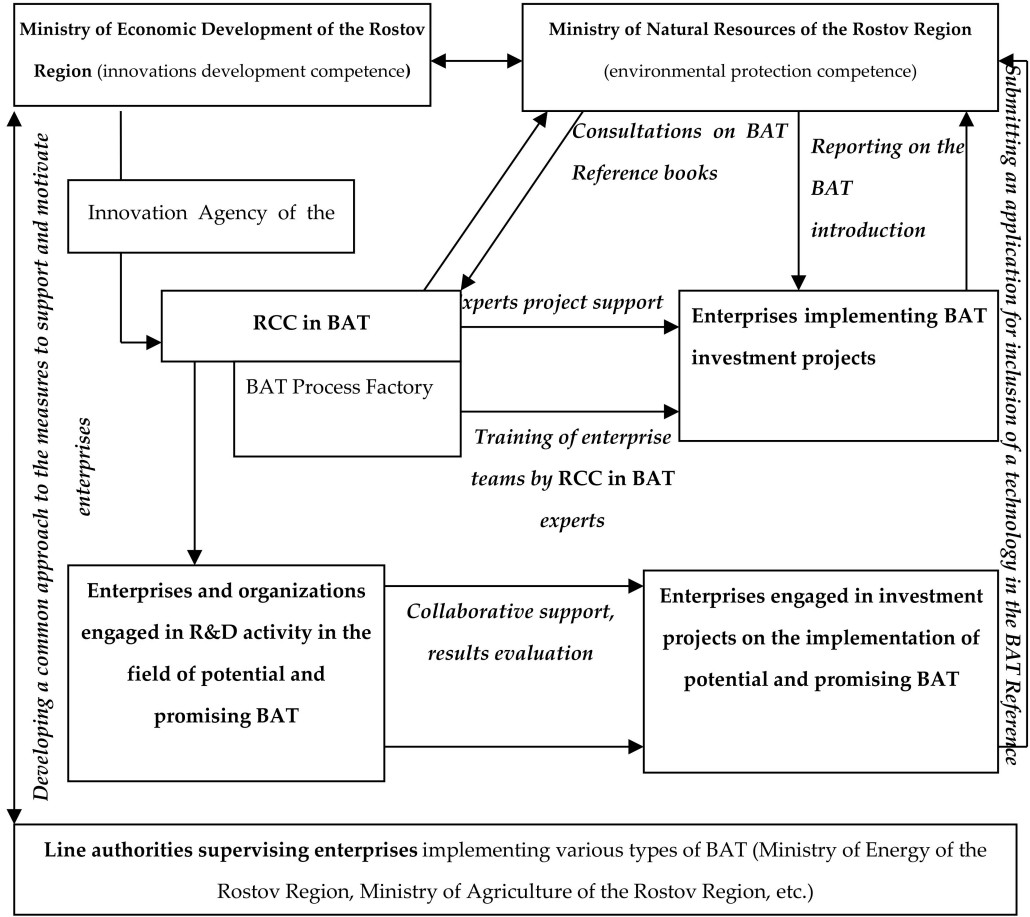

**Figure 2.** The place of the RCC in BAT in the general scheme of interaction of participants in the process of introducing various types of BAT in the region (compiled by the authors).

This structure can be created by analogy with the Regional Center of Competences in Labor Efficiency, already operating in the Russian regions within the framework of the national project "Labor Productivity", thus its successful experience in organizing their work would be helpful. RCC in BAT experts could provide free consulting services and project support for enterprises implementing programs to improve environmental efficiency with the introduction of existing BAT, as well as implementing developments in the field of potential and promising BAT. It is also be appropriate to include the creation of a "Factory of BAT processes" (also by analogy with the national project "Labor Productivity") under the RCC in BAT activities. It could become a place where the teams of such enterprises can be trained in tools for creating and implementing BAT for free [13].

RCC in BAT activities at the first stages (the first 4–5 years of the BAT implementation program in the region) are financed from the regional budget (if such a decision is made at the regional level). For enterprises introducing various types of BAT into their production process, the RCC in BAT experts' services should be free of charge, including support of investment projects for modernization based on various types of BAT and enterprise's staff teams training on gaining the experience and their further independent work on BAT implementation of at the enterprise. Partial self-financing could potentially be envisaged through fee-based services to enterprises that achieved satisfactory results from modernization on the basis of BAT in the first years.

An increase in the number of developed new technologies and BAT would meet the demand of local companies for innovative developments. So, innovation activity in the Rostov region is characterized by the following indicators (Table 2).

**Table 2.** Key indicators of innovation activity of the Rostov region.

| Indicator | 2017 | 2018 | 2019 | 2020 |
|---|---|---|---|---|
| The number of organizations engaged in research and development, units | 89 | 86 | 94 | 94 |
| The number of employees engaged in research and development, people | 11,846 | 11,720 | 11,974 | 11,940 |
| Internal operating costs for research and development, RUB bln | 12.7 | 12.6 | 14.5 | 13.3 |
| Capital costs for research and development, RUB mln | 382.90 | 341.70 | 1503.30 | 1310.80 |
| Share of Internal operating costs for research and development, % of gross regional product | 0.97 | 0.84 | 0.97 | no data |
| Costs for technological innovation, RUB bln | 28.25 | 19.59 | 36.55 | 52.71 |
| Volume of innovative goods, work, services, mln rubles | 104.5 | 64.5 | 62.7 | 106.7 |
| The percentage of innovative goods, work, services in the total volume of goods shipped, work performed, services, % | 10.6 | 5.8 | 4.9 | 8.5 |
| Patent applications for inventions (units) | 472 | 612 | 538 | 410 |
| Patents for inventions (units) have been issued | 634 | 447 | 494 | 447 |
| Advanced production technologies developed in the region, units | 15 | 19 | 17 | 16 |
| Advanced production technologies used in the region, units | 3368 | 3514 | 3872 | 4240 |

Source: Innovation infrastructure and key indicators of innovation activities of the members of the Russian Federation [compiled by the authors].

As the data presented in Table 2 show the Rostov region is marked with a significant—by two orders of magnitude—excess of the number of used advanced technologies in comparison with the developed ones. At the same time, the total number of developed technologies as a whole is approximately at the same level, and the number of technologies used is constantly growing. It proves that there is local enterprises' strong need and willingness to implement these changes.

The introduction into the practice of regional management of a mechanism based on the construction of the RCC on BAT will allow to obtain a dual effect:

—in terms of reducing the negative impact on the environment, modernizing and improving the energy efficiency of the production of the largest polluter enterprises through the introduction of BAT;

—in terms of improving the indicators of innovation activity in the region. In particular, the work of the RCC on BAT will directly affect to the greatest extent the number of advanced technologies developed in the region, the number of patents granted for inventions. To a lesser extent, this may affect the number of advanced technologies used, but there will be a replacement of foreign technologies used with domestic technologies. In general, this will contribute to the growth of the share of innovative goods, works, services in the total volume of goods shipped, works performed, services, as well as the volume of innovative goods, works, services in the region.

In the case of the development of a program-target document at the regional level for the introduction of BAT, it can be based on the following main target indicators and program activities (Table 3).

**Table 3.** The main possible targets and activities of the program for the introduction of BAT in the region [compiled by the authors].

| Indicator/event of the Program | Target Values for the Planning Period | | | | | |
|---|---|---|---|---|---|---|
| | 1st Year | 2nd Year | 3rd Year | 4th Year | 5th Year | 6th Year |
| The creation of the RCC on NDT, the formation of its staff and the training of experts, units, the final result is | 1 | 1 | 1 | 1 | 1 | 1 |
| The number of enterprises participating in the region participating in the program and implementing BAT, units, as a final result | - | 2 | 4 | 6 | 8 | . . . |
| Number of enterprises participating in the program and implementing promising and potential BAT, units, cumulative total | - | 2 | 4 | 6 | 8 | . . . |
| The number of innovative organizations participating in the program and carrying out developments in the field of potential and prospective BAT, units, increasing total | 2 | 4 | 6 | 8 | 10 | . . . |
| Number of patents for inventions issued to innovative organizations participating in the program, units. | - | 2 | 4 | 6 | 8 | . . . |
| The number of developed advanced production technologies in the region developed by innovative organizations participating in the program, units. | - | 2 | 4 | 6 | 8 | . . . |
| Creation of a "Factory of NDT processes", unit, increasing total | - | 1 | 1 | 1 | 1 | 1 |
| Development of a training program in the field of BAT for program participants | - | 1 | 1 | 1 | 1 | 1 |
| Number of trained employees of enterprises and innovative organizations participating in the program, people, cumulative total | - | 40 | 80 | 120 | 160 | . . . |

The program indicators are presented based on:

—the minimum staff of the RCC for NDT, including 8 people (including 1 head). The experts of the RCC on BAT can be industry specialists-technologists in the priority sectors for the enterprises of the region for the introduction of BAT. At the same time, 4 experts will be involved in working with enterprises implementing already existing BAT (based on approved BAT Reference Books), 4—with enterprises acting as platforms for the pilot implementation of promising and potential BAT based on the developments of regional innovative organizations;

—conducting by one expert of the RCC on BAT no more than two enterprises at the same time, and the requirements for supporting projects on the introduction of BAT at one enterprise simultaneously by two experts;

—the duration of projects on the introduction of BAT, potential and promising BAT in 3 years, the first two of which are accompanied directly by experts of the RCC on BAT in the region, the third is implemented by enterprises independently, with the fragmentary involvement of experts, if necessary;

—training from each company participating in a team of 10 people.

It is also assumed that in the first year of the program implementation, the recruitment of the RCC team on NDT will take place, they will develop training programs for future participating enterprises and innovative organizations.

If the proposed minimum indicators of the program are fulfilled, the number of developed advanced production technologies in the region will grow by about 50% over

the first five years of its implementation, while trends in all other factors (the general economic situation, measures to support innovative organizations, budget policy, etc.) remain unchanged.

The impact on other indicators of innovative development of the region, presented in Table 2, will be less significant and generally difficult to directly measure. For a more significant result at the regional level, it is necessary to increase the staff of experts by 2–3 times. In the case of a larger and longer-term development of events, the implemented measures will generally contribute to the improvement of almost all the indicators in Table 2, including the number of patents, the volume of innovative products and its share in the total volume of goods shipped, works performed, services. In general, a more precise determination of the impact of the implementation of the proposed measures on the future indicators of innovative development of the region requires further research.

Each of the regions of Russia specializes in its own innovative areas that can be used to develop BAT.

The priority areas of Rostov region specialization in potential BAT development, followed by their roll-out, are determined by the world-class Research and Educational Center of the Interregional Scientific and Educational Center of the South of Russia (hereinafter—the Southern REC), initiated by the Rostov and Volgograd regions and Krasnodar Territory. The Southern REC's list of activities includes three main areas: AgroTech, (creation of agricultural machinery and equipment for agriculture; creation of technologies for the production, storage and transportation of agricultural products, as well as reducing its losses, development of technologies for managing soil fertility and reducing the negative impact of man-made factors on the environment, rational waste management), FoodDesign (development of healthy food products, personalized nutrition, development of advanced gastronomy and culinary business, ensuring the adaptation of advanced food production technologies to the needs of consumers in the logic of preserving health and sustainability, as well as the creation of technologies and equipment for processing agricultural products, storage, transportation, packaging of food products), AquaTrack (development of improved methodologies for water resources management based on science and technology; introduction of new industrial intensive biotechnologies based on genetic research, disease prevention, targeted delivery of vaccines, new highly efficient, intelligent production management systems based on digital technologies; ensuring the sustainability of water resources (materials provided by the Ministry of Economic Development of the Rostov Region).

## 5. Conclusions

As priority areas for the development of BAT in the Rostov region (determined on the basis of the List of Activities of the Southern REC, according to materials provided by the Ministry of Economic Development of the Rostov Region), the following activities can be considered:

—transition to environmentally friendly and resource-saving energy, increase of hydrocarbon raw materials extraction and deep processing efficiency, development of new sources, energy's transportation and storage methods;

—transition to highly productive and environmentally friendly agricultural and aquaculture, development and implementation of systems for the rational use of chemical and biological protection of agricultural plants and animals.

Concentration of efforts on stimulating innovations in these areas, with their upgrade to the status of promising BAT followed by subsequent introduction to BAT Reference Books will provide innovative enterprises and organizations with strategic perspectives for going beyond the region and the country as a whole.

The development of Russian regional innovation systems based on the best available technologies can have a multiplier effect. This will not only ensure a reduction of negative environmental impact of large industrial and agro-industrial enterprises, but will also

contribute to the constant modernization and increase in the energy efficiency of production, as well as will enhance the innovation actors in the region.

The implementation of the proposed incentives for the establishment Regional BAT competence center will allow the Rostov region to join the overall process of BAT implementation at the federal level, as well as more effectively implement the Southern REC establishment policy. In the case of consistent implementation of the measures proposed, enterprises participating in this process will have opportunities to increase their export potential, while innovative organizations—to increase technological upgrading of innovative activities, reduce their risks and enter the latest technology markets both in other regions and abroad.

**Author Contributions:** Conceptualization, N.K.; Funding acquisition, S.T.; Investigation, M.P.; Methodology, K.S.; Resources, N.R. All authors have read and agreed to the published version of the manuscript.

**Funding:** The article was prepared with the financial support of the Russian Foundation for Basic Research within the framework of scientific project №. 19-010-00860 "Formation of an organizational and economic mechanism for managing the sustainable development of regional innovation systems based on the best available technologies." The research is partially funded by the Ministry of Science and Higher Education of the Russian Federation under the strategic academic leadership program 'Priority 2030' (Agreement 075-15-2021-1333 dated 30 September 2021).

**Institutional Review Board Statement:** Not Applicable.

**Informed Consent Statement:** Not Applicable.

**Data Availability Statement:** Not Applicable.

**Conflicts of Interest:** The authors declare no conflict of interest.

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
