# Peer review of "Development Priorities for the Regional Innovation System Based on the Best Available Technologies"

_sustainability, doi:10.3390/su14031116_

Round 1

Reviewer 1 Report

The article describes a regional mechanism for sustainable development of innovativeness based on best available technologies (BAT) in the Rostov region (Russia). Certain recommendations were formulated and a concept of a new institute for the implementation of BAT was proposed.

In the "Introduction" section certain statements focused on the benefits of BAT should be underpinned with some firm literature sources (other than only those of the co-authors). References based on other countries' cases would be beneficial. It should be made clear what Rosstandart is.

The literature review should be improved - a number of references are listed and assigned to certain topics, yet an in-depth analysis of their outcomes is not clearly visible. The Authors have identified their research gap mainly in reference to specific Russian regions, however, the comparative analysis with other regions (outside Russia) should be included. From a strictly scientific point of view, the knowledge gap is not convincing. The methodological approach is described in a very general and not convincing way. The research design, questions, hypotheses are not sufficiently presented.

In Table 2 key indicators of innovation activity of the Rostov region are presented - it is recommended to include information about the number of patents. A reliable forecast (e.g. based on analogy) on how a new center would influence these indicators would be beneficial. The expected impact of a new institution is not clearly visible in the manuscript (especially in a measurable way). 

Scientifically, the impact of the results coming from the manuscript and added value are not clearly visible. Research, practical and social implications are barely discussed.

General remarks

Editing (e.g. missing space, quality of figures), references (e.g. lack of information on access date when citing websites), and English should be improved.

Author Response

Good day! Thank you very much for your review. We have reviewed it and made adjustments in accordance with your comments. Thank you Good day! Thank you very much for your review. We have reviewed it and made adjustments in accordance with your comments. Thank you

Reviewer 2 Report

1) One of the factors ensuring the sustainable development of the innovation ecosystem (in this case, the ecosystem of the region) is the availability of developed technology transfer mechanisms. The model considered in the article is closed within the framework of describing the processes between actors in the same region, while some of the best available technologies (BAP) come from outside (including from abroad). It is advisable to consider the role of technology transfer centers and networks in the proposed model, especially taking into account the presence of such innovative structures in the region under consideration.
2) It is necessary to substantiate the possibility of using the developments obtained during the creation of the Regional Center of Competences in Labor Efficiency to build the Regional Center of Competence in BAT, since in the first case it is about optimizing business processes, and in the second - about the introduction of new technologies.
3) The terms AgroTech, FoodDesign, AquaTrack require explanation within the framework of the article
4) It is necessary to check the punctuation, use the same design of links, bring the drawings to a single style.

Author Response

Good day! Thank you very much for your review. We have reviewed it and made adjustments in accordance with your comments. Thank you

Round 2

Reviewer 1 Report

The paper has been significantly improved. However, there are still certain areas that need further amendments. 
Although the purpose of this article has been defined, the corresponding knowledge gap is not presented convincingly, due to the reason that very bearly no conclusions are retrieved from the literature review. At this point, the introduction and literature review presents more what has been done in certain corresponding areas, but it is not shown sufficiently what gaps exist in the literature that allowed this article to be written.
The paper lacks true discussion. It is recommended to discuss the results in line with previous literature. So far, it is only a presentation of results. No proper comparison with existing literature is made – neither in section „Results” nor in section „Conclusions”.

Author Response

Good day! Thanks for the comments, all adjustments have been corrected. Thank you

Reviewer 2 Report

I recommend bringing the drawings to a unified style (font of captions, placement of captions, connecting arrows, etc.)

Author Response

(The authors gave the same response as above.)
